# Missing the Point: Non-Convergence in Iterative Imputation Algorithms

Hanne I. Oberman [1]  Stef van Buuren [1] [2]  Gerko Vink [1]

## Abstract

Iterative imputation is a popular tool to accommodate missing data. While it is widely accepted that valid inferences can be obtained with this technique, these inferences all rely on algorithmic convergence. There is no consensus on how to evaluate the convergence properties of the method. Our study provides insight into identifying non-convergence in iterative imputation algorithms. We found that—in the cases considered—inferential validity was achieved after five to ten iterations, much earlier than indicated by diagnostic methods. We conclude that it never hurts to iterate longer, but such calculations hardly bring added value.

## 1. Iterative Imputation

To draw inference from incomplete data, most imputation software packages use iterative imputation procedures. With iterative imputation, the validity of the inference depends on the state-space of the algorithm at the final iteration. This introduces a potential threat to the validity of the imputations: What if the algorithm has not converged? Are the imputations then to be trusted? And can we rely on the inference obtained using the imputed data?

These remain open questions since the convergence properties of iterative imputation algorithms have not been systematically studied (Van Buuren, 2018, § 6.5.2). While there is no scientific consensus on how to evaluate the convergence of imputation algorithms (Zhu & Raghunathan, 2015; Takahashi, 2017), the current practice is to visually inspect imputations for

signs of non-convergence. This approach may be undesirable for several reasons: 1) it may be challenging to the untrained eye, 2) only severely pathological cases of non-convergence may be diagnosed, and 3) there is not an objective measure that quantifies convergence (Van Buuren, 2018, § 6.5.2). Therefore, a quantitative, diagnostic method to identify non-convergence would be preferred.

## 2. Identifying Non-Convergence

In our study, we consider two non-convergence identifiers: autocorrelation (conform Lynch, 2007, p. 147) and potential scale reduction factor $\widehat{R}$ (conform Vehtari et al., 2019, p. 5).[3] Aside from the usual parameters to monitor—chain means and chain variances (Van Buuren, 2018, § 4.5.6)—we also investigate convergence in the multivariate state-space of the algorithm.

We implement Van Buuren (2018, § 6.5.1)'s recommendation to track the parameter of scientific interest, and subsequently propose a novel parameter: the first eigenvalue of the variance-covariance matrix that is obtained after imputing the missing data. Compared to monitoring the estimate of scientific interest, this novel parameter has the appealing quality that it is not dependent on the complete-data inference. With that, it suits one of the main advantages of imputation techniques—solving the missing data problem and the substantive scientific problem separately.

We evaluate the performance and plausibility of these diagnostic methods through model-based simulation in R (R Core Team, 2020). For reasons of brevity, we only focus on the iterative imputation algorithm implemented in the popular `mice` package in R (Van Buuren & Groothuis-Oudshoorn, 2011).

## 3. Simulation Study

The aim of the simulation study is to determine the impact of non-convergence on the validity of statistical inferences, and to assess whether non-convergence

---

[*]Equal contribution  [1]Department of Methodology and Statistics, Utrecht University, Utrecht, The Netherlands [2]Netherlands Organisation for Applied Scientific Research TNO, Leiden, The Netherlands. Correspondence to: Hanne Oberman <h.i.oberman@uu.nl>.

Presented at the first Workshop on the Art of Learning with Missing Values (Artemiss) hosted by the $37^{th}$ International Conference on Machine Learning (ICML). Copyright 2020 by the author(s).

[3]As recommended by e.g. Cowles & Carlin (1996).

may be detected using several diagnostic methods. Inferential validity is reached when estimates are both unbiased and have nominal coverage across simulation repetitions ($n_{\text{sim}} = 1000$).

To induce non-convergence in the imputation algorithm we use two sets of simulation conditions: early stopping and missingness severity. Early stopping implies that we vary the number of iterations before terminating the imputation algorithm (between 1 and 100 iterations). The missingness severity is determined[4] by the proportion of incomplete cases (between 5 and 95%). We provide a summary of the simulation set-up in Algorithm 1, whereas the complete script and technical details are available from github.com/hanneoberman/MissingThePoint.

---

**Algorithm 1** Simulation set-up

    Simulate data
    repeat
       for all missingness conditions do
          Create missingness
          for all early stopping conditions do
             Impute missingness
             Perform analysis of scientific interest
             Compute non-convergence diagnostics
             Pool results across imputations
             Compute performance measures
          end for
       end for
       Combine outcomes of all conditions
    until all simulation repetitions are completed
    Aggregate outcomes across simulation runs

---

## 4. Results

Our results indicate that inferential validity is achieved after five to ten iterations, even under severe missingness conditions (i.e., up-to 95% of cases having missing values). Conditions with a lower proportion of incomplete cases yield valid inferences almost instantly—after just two iterations. This is in grave contrast to the non-convergence identified by the diagnostic methods. Autocorrelation- and $\widehat{R}$-values still indicate improving convergence after 20 to 30 iterations.

For example, in Figure 1 we show some results for a regression estimate. Depicted are percentage bias, coverage rate, autocorrelation and $\widehat{R}$ of the estimate. From this, we see that within a few iterations the percentage bias approaches zero and nominal coverage is quickly reached—even when autocorrelation and $\widehat{R}$ would still

signal non-convergence. This also holds for scenarios with severe missingness. Moreover, results seem invariant to the choice of parameter for the non-convergence diagnostics. The novel parameter that we propose has equal performance to the scientific estimate presented in Figure 1,[5] while having the advantage of being independent from the scientific model of interest.

## 5. Discussion

With this study, we show that iterative imputation algorithms can yield correct outcomes, even when a converged state has not yet formally been reached. Any further iterations would then burn computational resources without improving the statistical inferences. Preliminary findings suggest that these results also hold for more challenging scenarios, e.g. under different missingness mechanisms or imputation models.

Our study found that—in the cases considered— inferential validity was achieved after five to ten iterations, much earlier than indicated by the autocorrelation and $\widehat{R}$ diagnostics. Of course, it never hurts to iterate longer, but such calculations hardly bring added value.

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

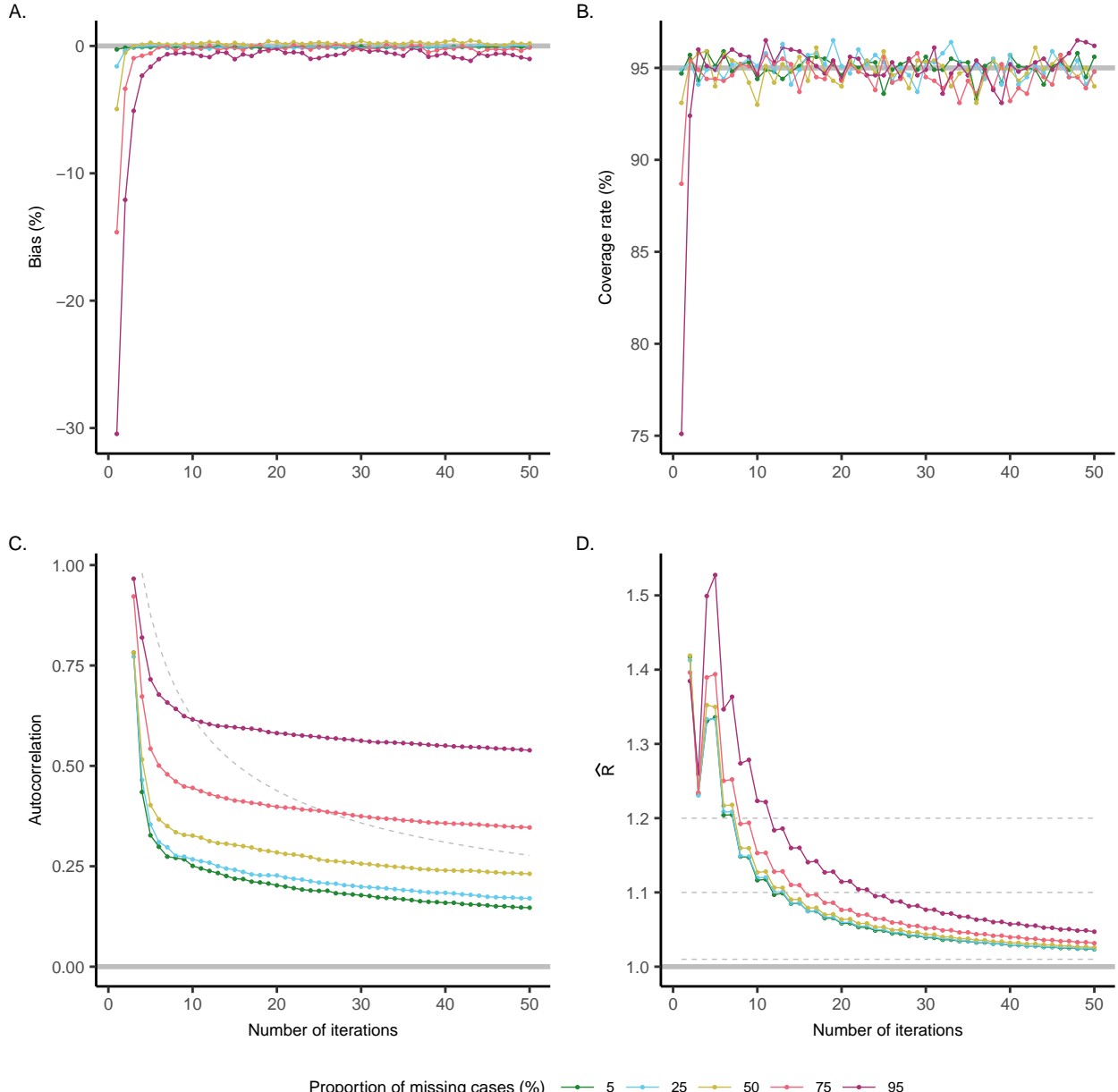

Figure 1. Subset of simulation results (truncated at 50 iterations). Depicted are performance measures percentage bias and coverage rate, and non-convergence identifiers autocorrelation and $\widehat{R}$ for a regression estimate. The solid gray lines represent inferential validity of the estimate (i.e., unbiasedness and nominal coverage), whereas the dashed gray lines depict common diagnostic thresholds of the identifiers.

R. Journal of Statistical Software, 45(1):1–67, December 2011. doi: 10.18637/jss.v045.i03.

Vehtari, A., Gelman, A., Simpson, D., Carpenter, B., and Bürkner, P.-C. Rank-normalization, folding, and localization: An improved $\widehat{R}$ for assessing convergence of MCMC. March 2019. URL http://arxiv.org/abs/1903.08008.

Zhu, J. and Raghunathan, T. E. Convergence Properties of a Sequential Regression Multiple Imputation Algorithm. Journal of the American Statistical Association, 110(511):1112–1124, July 2015. doi: 10.1080/01621459.2014.948117.
