# OpenReview forum: "Missing the Point: Non-Convergence in Iterative Imputation Algorithms"
_ICML.cc/2020/Workshop/Artemiss — ICML Artemiss 2020_

### Official Review · AnonReviewer2 · 2020-06-23
**convergence mice**

**Confidence:** 5
**Rating:** 7

**Review:**

Authors investigate the number of iterations required for multiple imputation by chained equations. The topic is relevant. Based on a simulation study, they mainly show using only 10 iterations can be sufficient for statistical analysis. In addition, they propose tools for diagnostic. In my opinion, additional investigations about model misspecification or categorical variables could be interesting.

---

### Decision · Program_Chairs · 2020-07-02

**Decision:**

Accept

**Comment:**

We are very happy to inform you that your paper has been accepted for the Artemiss workshop. We will contact you soon to inform you about the details concerning the format of your presentation at the workshop, and the camera-ready version deadline. Please take into account the referee's comments to write the camera-ready version.